# Lung Stereotactic Body Radiation Therapy in a Patient with Severe Lung Function Impairment Allowed by Gallium-68 Perfusion PET/CT Imaging: A Case Report

**DOI:** 10.3390/diagnostics13040718

**Published:** 2023-02-14

**Authors:** François Lucia, Mohamed Hamya, Fanny Pinot, David Bourhis, Pierre-Yves Le Roux

**Affiliations:** 1Radiation Oncology Department, University Hospital, 29200 Brest, France; 2LaTIM, INSERM, UMR 1101, University of Brest, 29200 Brest, France; 3Inserm, CHRU Brest, UMR 1304, GETBO, Nuclear Medicine Department, University of Brest, 29238 Brest, France

**Keywords:** Gallium-68 perfusion lung PET/CT, stereotactic radiotherapy, radiation-induced lung injury, very severe chronic obstructive pulmonary disease, case report

## Abstract

Lung stereotactic body radiotherapy (SBRT) is increasingly proposed, especially for patients with poor lung function who are not eligible for surgery. However, radiation-induced lung injury remains a significant treatment-related adverse event in these patients. Moreover, for patients with very severe COPD, we have very few data about the safety of SBRT for lung cancer. We present the case of a female with very severe chronic obstructive pulmonary disease (COPD) with a forced expiratory volume in one second (FEV1) of 0.23 L (11%), for whom a localized lung tumor was found. Lung SBRT was the only possible treatment. It was allowed and safely performed, based on a pre-therapeutic evaluation of regional lung function with Gallium-68 perfusion lung positron emission tomography combined with computed tomography (PET/CT). This is the first case report to highlight the potential use of a Gallium-68 perfusion PET/CT in order to safely select patients with very severe COPD who can benefit from SBRT.

**Figure 1 diagnostics-13-00718-f001:**
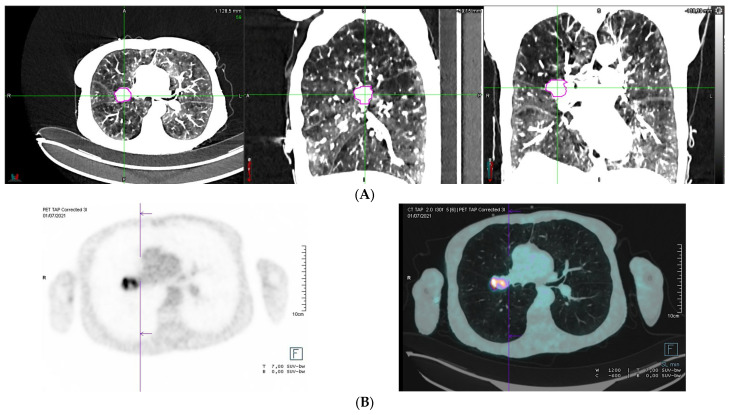
A 73-year-old female had a very severe chronic obstructive pulmonary disease (COPD) with a forced expiratory volume in one second (FEV1) of 0.23 L (11%). The patient received nocturn noninvasive ventilation and permanent oxygenotherapy. She was diagnosed with a lung tumor of the right hilum on chest radiography. On CT, a para-hilar mass of the posterior segment of the right upper lobe was measured to 29 mm × 23 mm in the axial plane (**A**). This mass was highly hypermetabolic on 18F-FDG PET/CT (**B**). The lesion was highly suspicious for a primary lung cancer. However, a biopsy was judged to be too risky. Multidisciplinary team meetings were proposed as the only possible therapeutic approach to treat this lesion by stereotactic body radiation therapy (SBRT) without histologic confirmation. Lung SBRT is considered to be a viable treatment option even in patients with poor lung function [1]. However, the reported incidence of symptomatic radiation-induced lung injury (grade ≥ 2) in the published data ranges from 9% to 28% [2,3,4,5] and is the most frequent adverse event after lung SBRT. Moreover, for patients with very severe COPD, we have very few data about the safety of SBRT for lung cancer [6,7]. Indeed, most clinical trials of SBRT for lung cancer have only included patients with FEV1 > 0.7 L [6,7]. Furthermore, retrospective studies about outcomes after SBRT in patients with severe COPD had very few patients with FEV1 < 0.7 L [8,9]. Therefore, for a patient with a FEV1 of 0.23 (11%), the therapeutic index was unknown. We decided to evaluate the possibility of treating this tumor with a SBRT protocol of 60 Gy in eight fractions due to the ultra-central localization [10]. This patient benefited from a perfusion PET/CT after the administration of ^68^Gallium-MAA to precisely evaluate lung function distribution and to assess the potential consequences of radiation therapy on lung function (Figure 2). Lung perfusion PET/CT images showed that a large part of the right lung was non-functional, especially the areas around the lung lesion. Lung functional volumes were delineated using a visually adapted semi-automatic method [11], and were co-registered with radiation plans. When we use this SBRT protocol, the dose constraints to both lungs are the anatomical lung volume receiving at least 26 Gy (V26 Gy) < 10% and the mean dose (Dmean) < 7 Gy [10]. However, these dose constraints have been evaluated in patients with FEV1 > 0.7 L and should be lower in patients with FEV1 < 0.7 L. In our case, the V26Gy (yellow line) to the anatomic lung volume was 4.7% versus only 1.1% to the functional lung volume. Furthermore, the Dmean to the anatomical lung volume was 5.29 Gy versus only 2.68 Gy to the functional lung volume. Based on this functional evaluation with functional lung volumes irradiated significantly below the limit values and in the absence of a therapeutic alternative, we approved the SBRT treatment. At 3 months of treatment, the CT scan showed a partial response to radiotherapy 15 vs. 29 mm (−49%) according to the Response Evaluation Criteria in Solid Tumors 1.1 (RECIST 1.1) (Figure 3) [12]. Toxicity was assessed using the Common Terminology Criteria for Adverse Events, version 5.0, with three questionnaires about quality of life (QLQ-C30, QLQ-LC13, and EQ-5D-5L) and with pulmonary function test. The patient experienced no side effects, and spirometry showed a FEV1 of 0.27 L (13%). Lung SBRT is the standard of care to treat inoperable stage 1 and 2 lung cancer [1]. However, the reported incidence of symptomatic radiation-induced lung injury (grade ≥ 2) in the published data ranges from 9% to 28% [1] and is the most frequent adverse event after lung SBRT. Moreover, most retrospective studies and ongoing clinical trials of SBRT for lung cancer have only included patients with a forced expiratory volume in one second (FEV1) > 0.7 L [6,8]. Therefore, for a patient with a FEV1 < 0.7 L, the therapeutic index is unknown. Patients with severe COPD frequently demonstrate high heterogeneity of regional lung function, with non-functional areas alternating with still functional regions, which should be preserved as much as possible. Lung perfusion PET/CT imaging is a novel imaging technique for regional lung function evaluation. Similar to conventional lung perfusion scintigraphy, images are obtained after intravenous administration of macro-aggregated albumin (MAA) particles, which embolize in the pulmonary capillaries according to pulmonary blood flow. However, as compared with lung scintigraphy, these molecules are not radiolabeled with Technetium-99m, but with Gallium-68, a ß+ isotope, allowing image acquisition with PET technology [13]. PET is an intrinsically superior technique for image acquisition, with greater sensitivity and better spatial and temporal resolutions [14,15]. This offers the opportunity to improve the accuracy of lung functional volumes delineation and its integration in thoracic radiation therapy planning. Indeed, unlike surgery, radiation therapy requires a more accurate assessment of the distribution of lung function because radiation doses can be given to both lungs, not just the tumor-affected lobe. Siva et al. demonstrated the feasibility of preserving lung functional volumes in a cohort of 14 patients treated with 3D conformal radiotherapy for NSCLC adapted to ^68^Ga-perfusion PET/CT [16]. However, SBRT is a more precise and conformal technique that allows the irradiation of smaller volumes of organs at risk (OARs) [17]. To date, no study has evaluated the performance of ^68^Ga-perfusion PET/CT imaging to personalize lung SBRT planning and dosimetry. In our case, the performance of lung perfusion PET/CT allowed a more accurate assessment of the doses delivered to the functional lung volume at the time of radiotherapy planning than lung perfusion scintigraphy. These data allowed us to improve the prediction of the risk of toxicity in this patient and to evaluate it as very low. This case report illustrates the potential use of a Gallium-68 perfusion PET/CT in order to safely select patients with very severe COPD who can benefit from SBRT.

**Figure 2 diagnostics-13-00718-f002:**
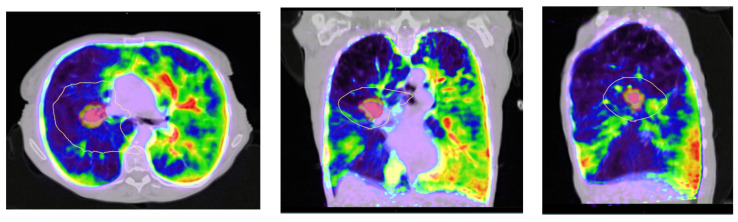
Axial (**left**), coronal (**middle**), and sagittal (**right**) slices of registration between lung perfusion PET/CT using Gallium-68-MAA (in red: the most functional areas; in blue: the least functional areas) and treatment planning computed tomography simulation scan. The planning target volume (PTV) is contoured in red; the V26Gy is the yellow line.

**Figure 3 diagnostics-13-00718-f003:**
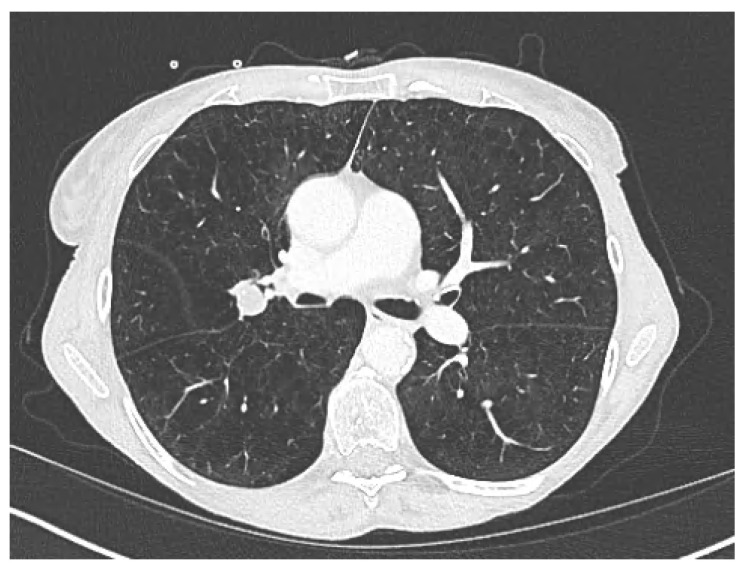
Computed tomography scan at 5 months of lung stereotactic radiotherapy.

## Data Availability

Research data are stored in an institutional repository and will be shared upon request to the corresponding author.

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
