# Peer review of "Lung Stereotactic Body Radiation Therapy in a Patient with Severe Lung Function Impairment Allowed by Gallium-68 Perfusion PET/CT Imaging: A Case Report"

_diagnostics, 2023, doi:10.3390/diagnostics13040718_

Round 1
Reviewer 1 Report
The authors provided a case report about a patient who received SBRT. The safety of SBRT in a patient with COPD was evaluated by 68Gallium perfusion PET/CT. This report is informative for SBRT prescribing. However, some points need to be corrected.
Abstract: “Lung stereotactic radiotherapy (SBRT)” should be “Lung stereotactic body radiotherapy (SBRT)”.
P2, L77. Is “the volume” the volume of functioning lungs?
P3, L92-95. The same sentence is repeated, please rephrase or summarize.
In Figure 2. Lung perfusion PET/CT using Gallium68-MAA showed functional lungs. Please specify the impact of this on the SBRT plan. Is it to avoid irradiating functional lungs?
Author Response
The authors provided a case report about a patient who received SBRT. The safety of SBRT in a patient with COPD was evaluated by 68Gallium perfusion PET/CT. This report is informative for SBRT prescribing. However, some points need to be corrected.
Abstract: “Lung stereotactic radiotherapy (SBRT)” should be “Lung stereotactic body radiotherapy (SBRT)”.
We have made the correction
P2, L77. Is “the volume” the volume of functioning lungs?
These are the recommendations for anatomical lung volume as there are no recommendations for functional lung volume. In our study, we used the same constraints in both cases.
“When we use this SBRT protocol, the dose constraints to the both lungs are the anatomical lung volume receiving at least 26 Gy (V26 Gy) < 10% and the mean dose (Dmean) < 7 Gy »
P3, L92-95. The same sentence is repeated, please rephrase or summarize.
We have summarized this part.
“Moreover, most of retrospective studies and ongoing clinical trials of SBRT for lung cancer only included patients with a forced expiratory volume in one second (FEV1) > 0.7 L [6,8]. »
In Figure 2. Lung perfusion PET/CT using Gallium68-MAA showed functional lungs. Please specify the impact of this on the SBRT plan. Is it to avoid irradiating functional lungs?
In this particular case, perfusion PET allowed us to accurately assess the doses delivered to the functional lung volume and to validate the SBRT in view of the very low doses given to the functional lung volume. Further sparing of the functional lung volume was not feasible due to the almost complete absence of functional lung volume in the treated area.
Reviewer 2 Report
This manuscript describes a case of a female with very severe chronic obstructive pulmonary disease (COPD) who was treatment with lung stereotactic radiotherapy (SBRT) entitle with “Lung stereotactic body radiation therapy in a patient with severe lung function impairment allowed by Gallium-68 perfusion PET/CT imaging: a case report”. The finding in this manuscript is interesting and would be helpful to the relevant research field. However, the manuscript is not well organized and written. For example, it was not written as a full article with introduction, results, experimental procedure and conclusion included, not a note, letter, or meeting abstract. This manuscript can be accepted after re-organized and re-written following the journal’s guideline.
Author Response
This manuscript describes a case of a female with very severe chronic obstructive pulmonary disease (COPD) who was treatment with lung stereotactic radiotherapy (SBRT) entitle with “Lung stereotactic body radiation therapy in a patient with severe lung function impairment allowed by Gallium-68 perfusion PET/CT imaging: a case report”. The finding in this manuscript is interesting and would be helpful to the relevant research field. However, the manuscript is not well organized and written. For example, it was not written as a full article with introduction, results, experimental procedure and conclusion included, not a note, letter, or meeting abstract. This manuscript can be accepted after re-organized and re-written following the journal’s guideline.
We are confused because we were asked by the editor to change our formatting to present it as the current version and not as an article as we had originally submitted. We don't know if we should change the formatting again
Round 2
Reviewer 2 Report
All my concerns have been well addressed, the manuscript can be accepted.